# Lateral Pectoral Nerve Identification through Ultrasound-Guided Methylene Blue Injection during Selective Peripheral Neurectomy for Shoulder Spasticity: Proposal for a New Procedure

**DOI:** 10.3390/jpm14010116

**Published:** 2024-01-20

**Authors:** Paolo Zerbinati, Jonathan Bemporad, Andrea Massimiani, Edoardo Bianchini, Davide Mazzoli, Davide Glorioso, Giuseppe della Vecchia, Antonio De Luca, Paolo De Blasiis

**Affiliations:** 1Neuro-Orthopedic Unit, Sol et Salus Hospital, 47922 Rimini, Italyj.bemporad@soletsalus.com (J.B.); d.glorioso@soletsalus.com (D.G.); 2Department of Neuroscience, Mental Health and Sensory Organs (NESMOS), Sapienza University of Rome, 00189 Rome, Italy; andrea.massimiani@uniroma.it (A.M.); edoardo.bianchini@uniroma1.it (E.B.); 3Gait and Motion Analysis Laboratory, Sol et Salus Hospital, 47992 Rimini, Italy; d.mazzoli@soletsalus.com; 4Department of Women, Child, General and Specialistic Surgery, University of Campania “L. Vanvitelli”, 80138 Naples, Italy; dellavecchia-g@libero.it; 5Section of Human Anatomy, Department of Mental and Physical Health and Preventive Medicine, University of Campania “Luigi Vanvitelli”, Via Luciano Armanni, 5, 80138 Naples, Italy; antonio.deluca@unicampania.it

**Keywords:** spasticity, shoulder spasticity, selective neurectomy, functional surgery, lateral pectoral nerve, ultrasonography, methylene blue

## Abstract

Internally rotated and adducted shoulder is a common posture in upper limb spasticity. Selective peripheral neurectomy is a useful and viable surgical technique to ameliorate spasticity, and the lateral pectoral nerve (LPN) could be a potential good target to manage shoulder spasticity presenting with internal rotation. However, there are some limitations related to this procedure, such as potential anatomical variability and the necessity of intraoperative surgical exploration to identify the target nerve requiring wide surgical incisions. This could result in higher post-surgical discomfort for the patient. Therefore, the aim of our study was to describe a modification of the traditional selective peripheral neurectomy procedure of the LPN through the perioperative ultrasound-guided marking of the target nerve with methylene blue. The details of the localization and marking procedure are described, as well as the surgical technique of peripheral selective neurectomy and the potential advantages in terms of nerve localization, surgical precision and patients’ post-surgical discomfort. We suggest that the proposed modified procedure could be a valid technique to address some current limitations and move the surgical treatment of spasticity toward increasingly tailored management due to the ease of nerve identification, the possibility of handling potential anatomical variability and the resulting smaller surgical incisions.

## 1. Introduction

Spasticity is a common and complex motor phenomenon following upper motor neuron injury [1] characterized by muscle hyperactivity, with velocity-dependent hypertonia and abnormally increased tendon jerks [2]. Among involved body segments, upper limb spasticity (ULS) is common in post-stroke patients [3], with over 40% of patients reporting it [4,5], and leads to a potentially high functional limitation for patients [6,7]. Several pattern and postures have been described for ULS [8] and the involvement of the shoulder, in particular with internally rotated and adducted arm posture, is present in the vast majority of patients presenting with ULS [8]. Moreover, from 8% to 13% patients with post-stroke spasticity suffer from shoulder pain, a percentage that increases to over 25% in presence of disabling spasticity [9].

The severity of spasticity can vary greatly and, over time, this condition can lead to retractions, contractures, deformities, pressure ulcers and skin maceration [2], with a high impact on patients’ quality of life [10,11]. Therefore, the prompt identification and effective management of spasticity is crucial to assuring the best possible outcome for patients. To this end, several treatment options are available, including pharmacological therapy, rehabilitative treatment, chemodenervation techniques (e.g., botulinum toxin injection, nerve blocks) and surgery [12]. Among this latter option, possible strategies for spasticity management include selective and hyperselective peripheral neurectomies [13].

Peripheral neurectomies consist of the partial excision of the fibers of a motor nerve innervating spastic muscles [14]. The first one was described by Stoffel and colleagues [15] and then by Brunelli and Brunelli [16], becoming increasingly common procedures. While their origins date back to the first decades of the 20th century, selective peripheral neurectomy has received growing attention in recent years since this procedure has been demonstrated to mitigate muscle spasticity, particularly in lower limbs [17,18,19,20,21]. Based on this evidence, in fact, an international, interdisciplinary Delphi panel recently included selective neurectomies among the treatments for poststroke equinovarus foot deformities [22].

Although no recommendations exist for upper limbs, several studies demonstrated the effectiveness of selective peripheral neurectomy in ULS as well [13,14,18,23,24,25,26,27]. Regarding shoulder spasticity, only very few papers exist. In one study from Lin and colleagues [23], hyperselective neurectomy of thoracodorsal nerve was demonstrated to be effective in the treatment of shoulder spasticity. In another retrospective study, Sitthinamsuwan and collaborators [27] demonstrated the efficacy of selective neurotomy in the treatment of refractory ULS, including 14 patients undergoing pectoral nerve neurotomy. Lateral pectoral nerve (LPN) is a good potential target when surgically treating shoulder spasticity presenting with internal rotation. This nerve, indeed, innervates the pectoralis major muscle [28], one of the primary targets in shoulder spasticity management using botulinum toxin [29]. Moreover, diagnostic LPN nerve block has recently been proposed as part of an algorithm to evaluate hemiplegic shoulder pain [30].

During selective peripheral neurectomy of the LPN, after surgical incision in the infraclavicular fossa and partial reflection of clavipectoral fascia and pectoralis major, surgical exploration is carried out, and the nerve is located using electrical stimulation while observing the contraction of the pectoralis major. After locating the nerve motor fibers, a partial section is performed [27].

However, there are some limitations related to this procedure: (i) potential anatomical variability [31,32,33], which could increase surgical operative time and difficulty in locating the nerve; (ii) the fact that in order to perform surgical exploration, a sufficiently wide surgical incision is necessary; and (iii) the necessity of intraoperative search and identification of the nerve. This could also result in higher post-surgical discomfort for the patient.

## 2. Materials and Methods

The aim of this paper is to describe a modification of the traditional selective peripheral neurectomy procedure of the LPN through the perioperative ultrasound-guided marking of the target nerve with methylene blue. The details of the localization and marking procedure are described, as well as the surgical technique of peripheral selective neurectomy and the potential advantages in terms of nerve localization, surgical precision and patients’ post-surgical discomfort.

## 3. Results

### 3.1. Patient Selection and Indications

Patients with shoulder spasticity presenting with internally rotated and adducted arm posture could have an indication to undergo the selective peripheral neurectomy of the LPN described in this paper. The proposed technique is primarily described for adult patients with acquired spasticity; however, this could, in principle, be extended to pediatric patients as well since no contraindication to the use of methylene blue is present in this population and selective neurectomies are performed in children to treat ULS [24]. An assessment of patient motivation and cognition could be performed during each patient’s evaluation. However, since several conditions leading to spasticity could also involve cognitive impairment, cognitive deficit should not be considered an absolute contraindication to the procedure as long as the patient can cope with a surgical procedure and the postoperative regimen. A goal attainment scale could be included in each patient’s evaluation in order to guarantee a patient-centric approach. Since selective peripheral neurectomy is only effective on spasticity and has no effect on muscle or joint contractures, if the presence of a structural alteration such as those previously mentioned is suspected, a diagnostic nerve block of the LPN might be performed in order to predict the outcome of the surgical procedure. This is similar to the recommendation for the management of poststroke equinovarus foot deformities proposed by Salga and colleagues [22]. In the case of limited or no response to the diagnostic nerve block, other management strategies, such as functional surgery (e.g., tendon elongment, tenothomies, etc.) could be taken into consideration. Known hypersensitivity to methylene blue is a contraindication to this procedure and should lead to the choice of an alternative technique. General contraindications for surgery in patients with ULS linked to the procedure or anesthesia also apply to this technique, such as dystonia or other movement disorders, a lack of patient compliance, unrealistic expectations or high intraoperative risk, as judged by an anesthesiologist and surgeon.

### 3.2. US-Guided Nerve Localization and Marking

A sonographic unit coupled with a linear multifrequency probe is used for ultrasonographic guidance. With the patient in a supine position, the operator positions themselves on the same side as the target muscle. The LPN is located at the brachial plexus origin in the subclavian region, laterally to the hemiclavear line and approximately 3 cm below the clavicle, by positioning the probe parallel to the direction of the pectoralis major fibers. The nerve is then followed in the cranio-caudal direction, between the pectoralis major and pectoralis minor muscles, until the motor fibers for the pectoralis major muscle are identified (Figure 1).

The exact point at which the motor fibers detach from the main nerve trunk can be easily identified by performing slow cranio-caudal and caudo-cranial probe movements following the nerve course. This is crucial since the branching point is the site at which the neurectomy will be carried out.

After identifying the motor fiber origin location, a needle electrode connected to an electro-stimulator is inserted, in aseptic conditions, under ultrasound guidance and positioned in the proximity of the nerve previously identified. A current of 0.8–1 mA at the nerve site is then used to confirm the identified nerve by eliciting a contraction in the pectoralis major muscle. Once an adequate muscle contraction is evoked, the intensity of the stimulus is progressively reduced until the minimum possible stimulus capable of determining visible muscle contraction is reached (usually around 0.2–0.4 mA).

Thus, 0.8–1.2 mL of methylene blue solution 1% (Figure 2) is injected around the target nerve (Figure 3).

### 3.3. Setup

Selective neurectomy is performed with the patient in a supine position under general anesthesia. Paralysis is contraindicated because of the need for muscle contraction following intraoperative nerve stimulation.

### 3.4. Exposure

After general anesthesia, a transverse lateral subclavicular incision of approximately 2 cm is made at the site at which the branching point was identified, using the needle entrance point as a guide. It is usually 2–3 cm below the lateral third of the clavicula but can vary from individual to individual. Hemostasis is performed, and partial reflection of the great pectoralis is carried out, exposing the pectoralis minor and the LPN lying in the plane between the two muscles. The nerve branch can be easily identified at this level by the methylene blue marking. A silicone elastic loop is then placed around the nerve to isolate it (Figure 4).

An intraoperative electrostimulation is then used to confirm the identified nerve (Figure 5) by observing the contraction of the pectoralis major muscle.

### 3.5. Selective Peripheral Neurectomy

Once the isolated nerve is confirmed to comprise the target motor fibers of the LPN, the selective peripheral neurectomy is carried out by performing a microsurgical partial section of the motor nerve fibers. The epineurium is incised along the long axis of the nerve, and 50 to 75% of all fascicles are resected from the main trunk, depending on the extent of spasticity and the desired outcome (Figure 6).

Coagulation of the proximal and distal stumps is performed in order to prevent nerve regrowth and to slow sprouting.

### 3.6. Closure and Postoperative Care

Once the neurectomy is completed, the surgical wound is sutured and the procedure is concluded. Postoperative care consists of a soft, nonadherent dressing until the wound is healed. Gentle passive stretching and active exercises of the involved muscles are subsequently initiated according to patient conditions.

## 4. Discussion

The aim of this paper is to describe a modification to the traditional selective peripheral neurectomy procedure of the LPN through the perioperative ultrasound-guided marking of the target nerve with methylene blue.

The use of elective peripheral neurectomy to mitigate muscle spasticity has received growing attention in recent years. While most evidence supported its use in lower limbs [17,18,19,20,21] and a recently developed recommendation included selective neurectomies among treatments for post-stroke equinovarus foot deformities [22], no recommendations exist for upper limbs. However the effectiveness of selective peripheral neurectomies to treat ULS has been reported [13,14,18,23,24,25,26,27]. In our proposed technique, 50–75% of nerve fascicles are excised during the selective peripheral neurectomy. There is no total agreement on what percentage of the nerve needs to be resected, and this has critical implications for the recurrences that can occur several months after surgery. Sprouting from adjacent axonal endings has been linked to reinnervation and, for this reason, several authors recommended the excision of a relevant percentage of fascicles. Brunelli and Brunelli preformed the excision of around two-thirds of fascicles [16], Maarrawi et al. described the resection of 50% to 80% of the isolated motor branches paired with proximal coagulation to prevent regrowth [34]; Puligopu and Purhoit described the resection of one-third to three-quarters of the motor branches for individual muscles [35]. Similarly, Fouad reported the resection of 50% to 80% of the isolated motor branches of fascicles [36], and, finally, Leclercq described the excision of approximately two-thirds of each nerve branch at the level of the motor branches [24].

It emerges that a detailed knowledge of anatomy is essential in order to plan the operative act; therefore, the literature stresses the concept that guidance techniques and methods to improve nerve branch identification during dissection are crucial [37].

Our work falls into this line of research by using methylene blue to mark target nerve branches during selective peripheral neurectomy. The use of methylene blue as a surgical marker is not new. In fact, previous studies reported its use as a surgical marker to identify parathyroid glands [38] and during lung [39] and abdominal surgery [40]. Moreover, in our experience, the use of methylene blue did not constitute an obstacle to the surgeon’s view due to tissue soiling, and reported local adverse events linked to methylene blue toxicity are extremely rare and usually follow more demolitive surgeries such as breast and colon surgeries using higher doses and different dilutions [41].

Several advantages could arise from the proposed nerve marking. The first advantage of the US-guided perioperative identification and marking of the LPN is handling potential anatomical variability. Indeed, the LPN arises most frequently with two branches from the anterior divisions of the upper and middle plexus trunks. However, although anatomical variability seems to be less than in other districts, some variations are described, such as emergence as a single root from the lateral cord or emergence only from middle or upper trunks [37,38,39]. Anatomical variability could result in difficulties in reliably locating the nerve, increases in surgical operating time and potentially greater post-surgical discomfort for patients. To this end, this procedure has the potential to overcome this limitation since the LPN is easily identified through methylene blue colorization.

Secondly, the ease of nerve identification and the possibility of using the needle entrance hole as an anatomical landmark could also limit the need for surgical exploration, thus reducing the necessity of wider incisions and longer surgical operative time. Figure 7 shows a comparison of surgical incisions in both traditional and modified LPN selective peripheral neurectomy procedures. A smaller surgical wound will likely result in reduced patient discomfort, faster recovery and the reduced occurrence of postoperative complications.

Finally, it is important to underline that the addition of methylene blue marking will hardly affect the selective peripheral neurectomy procedure’s time and complexity. Indeed, as discussed above, the procedure will likely lead to a reduction in operative time and greater surgical efficiency. Moreover, nerve identification thorough electrical stimulation is routinely performed in the classical procedure, and the marking phase only adds the injection step, which does not require specific expertise or equipment other than a needle electrode suitable for injection and the methylene blue solution.

Given all these considerations, the proposed modified LPN selective peripheral neurectomy with the use of a perioperative US-guided methylene blue injection could be a valid technique to address some current limitations and move the surgical treatment of spasticity toward increasingly tailored and personalized management. Moreover, this technique can be applied to other districts in which anatomical variability could be a more difficult aspect to address. Future studies are warranted to evaluate the efficacy and the safety profile of this procedure and compare it to the traditional technique.

## Figures and Tables

**Figure 1 jpm-14-00116-f001:**
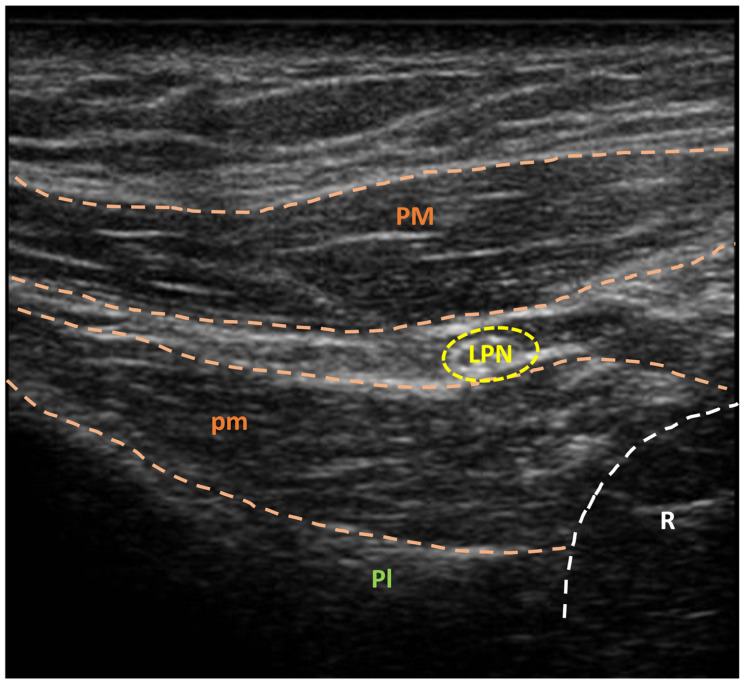
Ultrasonographic visualization of lateral pectoral nerve (yellow circle) before methylene blue injection. LPN: lateral pectoral nerve; Pl: pleura; PM: pectoralis major muscle; pm: pectoralis minor muscle; R: rib. A sonographic unit (Sonoscape X3, Sonoscape Europe s.r.l., Rome, Italy), coupled with a linear multifrequency probe (5–14 MHz, L741, Sonoscape Europe s.r.l., Rome, Italy), was used for ultrasonographic guidance.

**Figure 2 jpm-14-00116-f002:**
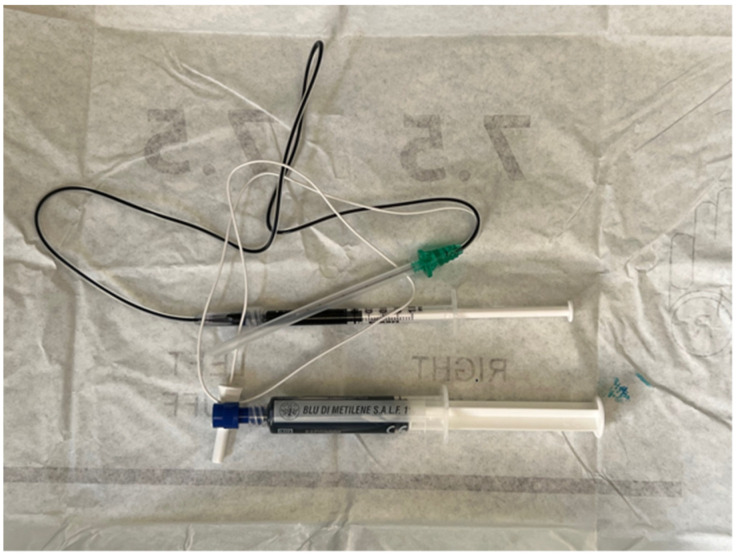
Syringe containing methylene blue solution 1% connected to a needle electrode.

**Figure 3 jpm-14-00116-f003:**
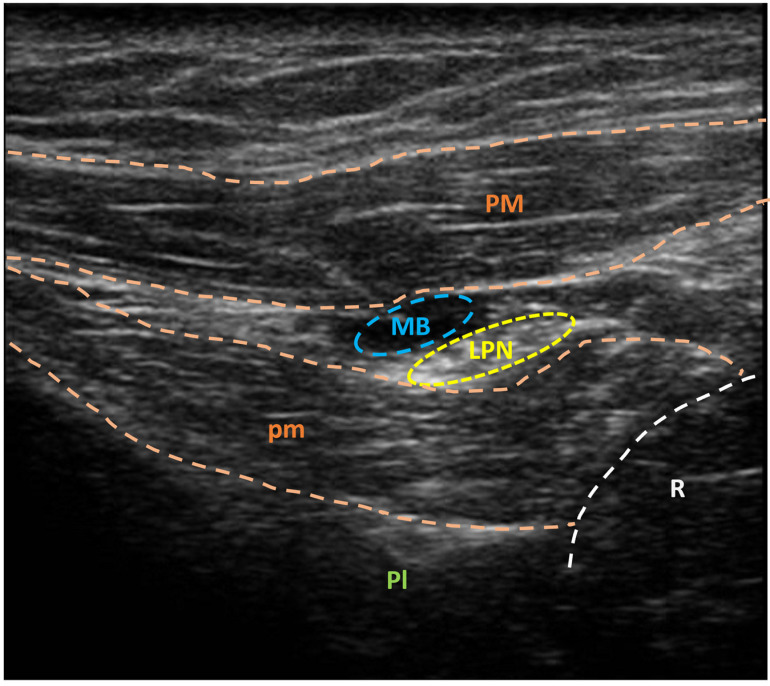
Ultrasonographic visualization of lateral pectoral nerve (yellow circle) after methylene blue injection (blue circle). LPN: lateral pectoral nerve; MB: methylene blue; Pl: pleura; PM: pectoralis major muscle; pm: pectoralis minor muscle; R: rib. A sonographic unit (Sonoscape X3, Sonoscape Europe s.r.l., Rome, Italy), coupled with a linear multifrequency probe (5–14 MHz, L741, Sonoscape Europe s.r.l., Rome, Italy), was used for ultrasonographic guidance.

**Figure 4 jpm-14-00116-f004:**
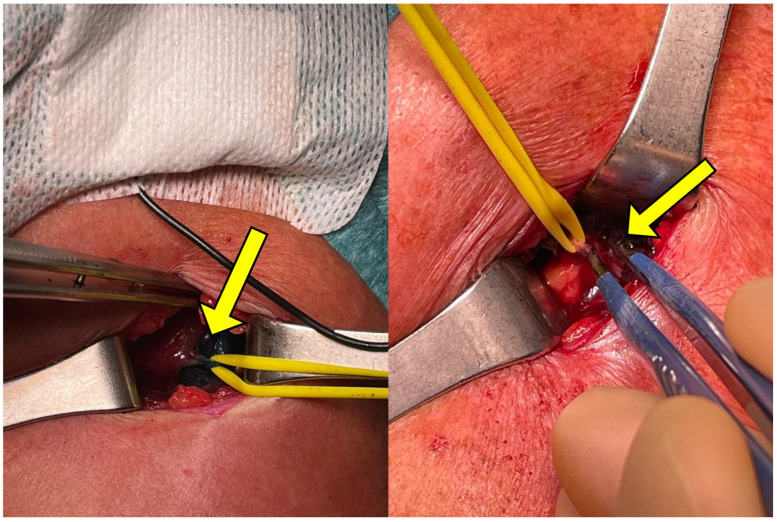
LPN visualization during surgical procedure with silicone loop placed for nerve isolation. The target nerve is located through the methylene blue marking colorization (arrow).

**Figure 5 jpm-14-00116-f005:**
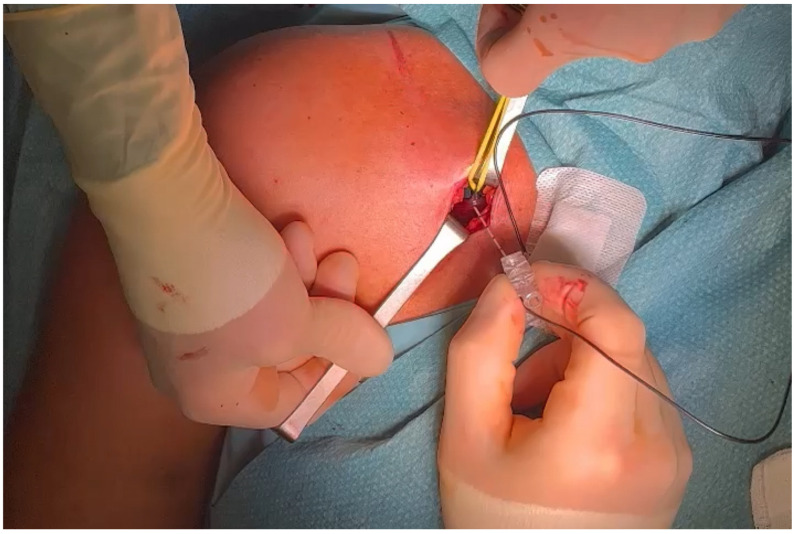
Electrostimulation during surgical procedure to confirm the identified nerve using methylene blue marking.

**Figure 6 jpm-14-00116-f006:**
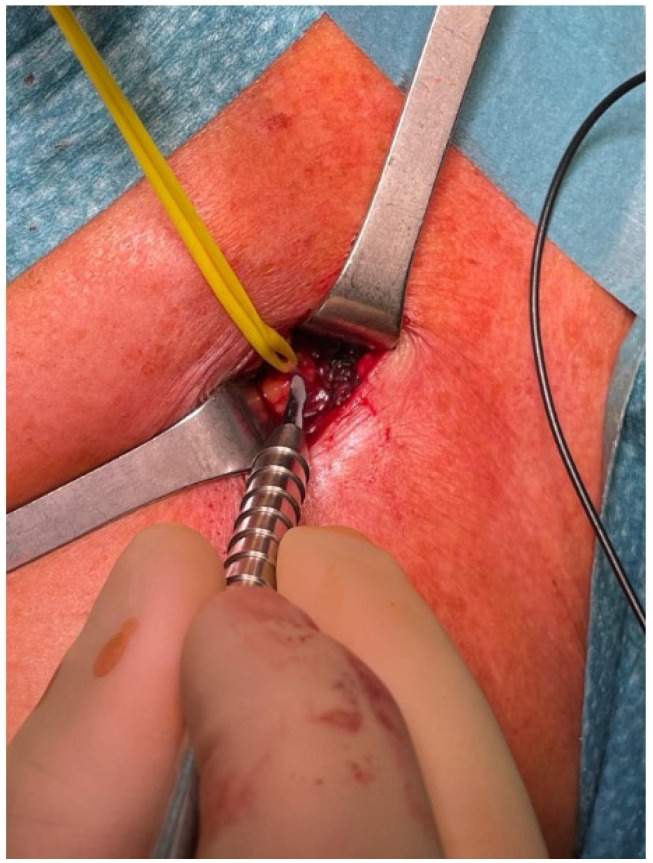
Fiber partial section of the LPN marked with methylene blue during selective neurectomy procedure.

**Figure 7 jpm-14-00116-f007:**
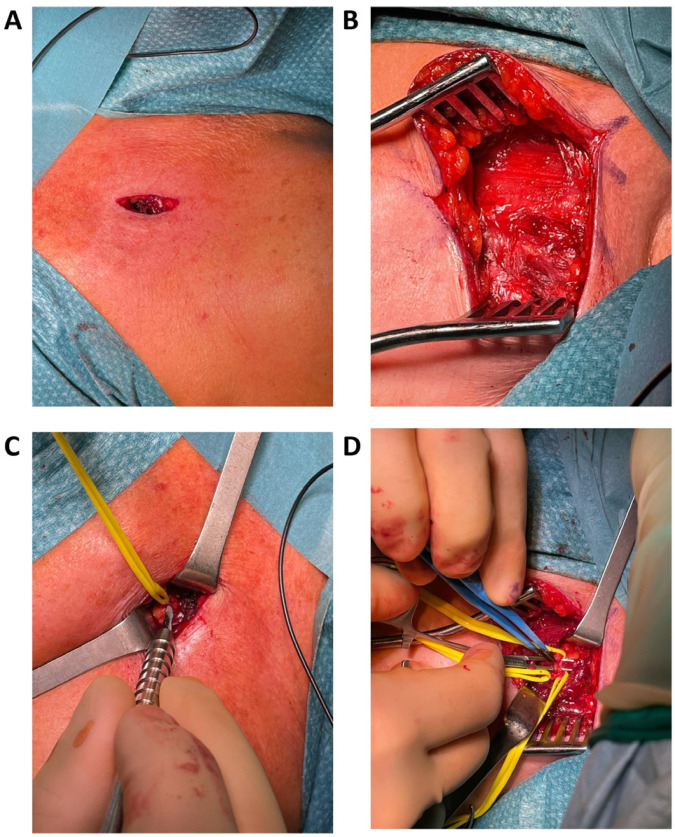
Comparison of surgical incision and intraoperative surgical wound in the proposed procedure with methylene blue injection (**A**,**C**) and in a traditional LPN neurectomy (**B**,**D**).

## Data Availability

Not applicable. A novel procedure is described and there is no data.

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
