# Peer review of "Lateral Pectoral Nerve Identification through Ultrasound-Guided Methylene Blue Injection during Selective Peripheral Neurectomy for Shoulder Spasticity: Proposal for a New Procedure"

_jpm, 2024, doi:10.3390/jpm14010116_

Round 1

Reviewer 1 Report

Comments and Suggestions for Authors
the authors described a modification of the traditional selective peripheral neurectomy procedure of LPN through perioperative ultrasound-guided marking of the target nerve with methylene blue. this kind of modification can bring tailored and personalized treatment to individual patients. this technology is really valuable and worth to be introduced extensively. if the number of the cases is larger, the comparison and following results will be more convincing. 

Author Response

The authors described a modification of the traditional selective peripheral neurectomy procedure of LPN through perioperative ultrasound-guided marking of the target nerve with methylene blue. this kind of modification can bring tailored and personalized treatment to individual patients. this technology is really valuable and worth to be introduced extensively. If the number of the cases is larger, the comparison and following results will be more convincing.

R: We thank the reviewer for the valuable comment. In the present work we aimed at describing the modified procedure exclusively. We are planning to present data on large number of our cases in order to show other reached endpoints in future works.

Reviewer 2 Report

Comments and Suggestions for Authors

This paper describes a proposed technique for lateral pectoral nerve identification with the aid of U/S and methylene blue aiming to a minimal incision for selective neurectomy. There is a paucity of literature dealing with acquired spasticity of the upper limb, so this new technique is interesting.  

I would like to make some remarks for the authors, and if adequately answered I would promote this article for publication: 

·      In my opinion the description of a surgical technique should be better organized in sections (relevant anatomy; indications; contraindications; US-guided nerve localization and marking; setup; exposure; selective peripheral neurectomy; closure; postoperative care and rehabilitation). All relevant information of each sub-section is valuable for the reader.

·      In the patients’ selection section, more info is necessary. Is acquired spasticity of the adults the only indication? Is spasticity due to cerebral palsy in children also an indication? 

·      In the case of acquired spasticity is there a time delay of surgical intervention so that the patient achieves maximal spontaneous motor recovery? Is the delay different for stroke or traumatic brain injury? 

·      Do you screen patients for cognitive deficits, motivation, and body image awareness when choosing to proceed with this procedure?

·      What about the contractures? I assume the existence of a contracture is a contraindication for this procedure. Please report the contraindications and discuss the management of the contractures in the discussion section (i.e tendon lengthening, humeral osteotomy).

·      Do you perform an excision of the nerve fascicles to avoid recurrence?

·      A more extended discussion section would be desirable. Where the technique stands in the context of the literature and implications for clinical practice and future studies should be analyzed in the discussion. I. think that some information from the introduction section better fits the discussion section.

·      Do you perform this technique as an adjunct to a standard functional rehabilitation program?

Author Response

This paper describes a proposed technique for lateral pectoral nerve identification with the aid of U/S and methylene blue aiming to a minimal incision for selective neurectomy. There is a paucity of literature dealing with acquired spasticity of the upper limb, so this new technique is interesting.

I would like to make some remarks for the authors, and if adequately answered I would promote this article for publication:

Q1: In my opinion the description of a surgical technique should be better organized in sections (relevant anatomy; indications; contraindications; US-guided nerve localization and marking; setup; exposure; selective peripheral neurectomy; closure; postoperative care and rehabilitation). All relevant information of each sub-section is valuable for the reader.

R1: We thank the reviewer for the valuable suggestion. We reorganized the procedure description according to reviewer’s indications.

Q2: In the patients’ selection section, more info is necessary. Is acquired spasticity of the adults the only indication? Is spasticity due to cerebral palsy in children also an indication?

R2: We thank the reviewer for the comment. We included a paragraph in the results section specifying that the proposed technique is primarily described for adult patients with acquired spasticity; however, this could be in principle extended to pediatric patients as well since no contra-indication to the use of methylene blue is present in this population and selective neurectomies are performed in children to treat Upper Limb Spasticity.

Q3: In the case of acquired spasticity is there a time delay of surgical intervention so that the patient achieves maximal spontaneous motor recovery? Is the delay different for stroke or traumatic brain injury?

R3. We thank the reviewer for pointing this out. Timing and prediction of maximal recovery have not been completely elucidated in literature. Indeed, some authors proposed to assess the residual potential benefit from neurectomy by performing a diagnostic nerve block (See, for reference, Fitterer et al. 2021, doi:10.3389/fneur.2021.668370). Moreover, it must be considered that spasticity management could not aim at restoring motor function, but at improving posture, prevent secondary complications and contractures, facilitating hygiene, or reduce pain. However, since the main aim of our work was to describe the procedure exclusively, we considered an extended discussion on spasticity timing and general management beyond the scope of our paper.

Q4: Do you screen patients for cognitive deficits, motivation, and body image awareness when choosing to proceed with this procedure?

R4: We thank the reviewer for the comment. We do not perform routinely an assessment of body image awareness. An assessment of patient motivation and cognition could be performed during patient’s evaluation. However, since several conditions leading to spasticity could also involve cognitive impairment, cognitive deficit should not be considered as an absolute contraindication to procedure. We included a paragraph in the results section on this aspect.

Q6: What about the contractures? I assume the existence of a contracture is a contraindication for this procedure. Please report the contraindications and discuss the management of the contractures in the discussion section (i.e tendon lengthening, humeral osteotomy).

R6: We thank the reviewer for pointing this out. We expanded the statement on the use of diagnostic nerve block for the prediction of attainable outcome and potential alternative strategies in the results section.

Q7: Do you perform an excision of the nerve fascicles to avoid recurrence?

R7: We thank the reviewer for the comment. We performed a neurectomy and not a neurotomy, therefore the excision of a certain percentage of fascicles (50%-75%) was carried out. We clarified this aspect in the procedure description.

Q8: A more extended discussion section would be desirable. Where the technique stands in the context of the literature and implications for clinical practice and future studies should be analyzed in the discussion. I. think that some information from the introduction section better fits the discussion section.

R8: We thank the reviewer for the suggestion. We expanded the discussion as suggested and moved some information from introduction to discussion.

Q9: Do you perform this technique as an adjunct to a standard functional rehabilitation program?

R9: We thank the reviewer for the comment. The patients undergoing the described procedure enroll in rehabilitation. However, the program is not specific for people undergoing this kind of modified procedure. We did not report information on post-surgical management since it is not linked to the specific technique and we wanted to focus on procedure description.

Reviewer 3 Report

Comments and Suggestions for Authors

The authors presented modification of selective peripheral neurectomy of LPN. The manuscript is interesting and has merit. I only have a few comments:

1. Can the authors provide a better US image of the point where motor branch emerges from LPN? Also please mark other anatomical structures in the US image and add some details about US machine/probe in the Figure caption.

2. How many times did authors perform this procedure? If this is a case report, the language should be written in past tens. 

3. Are there any limiations of using methylene blue? Can it obstruct the view of a surgen if strcutures are coloured blue? In our experience using 0,5% methylene blue, it can soil the tissue quite significantly, obstructing the clear view of anatomy. Can it be potentially locally toxic? 

Author Response

Reviewer #3

The authors presented modification of selective peripheral neurectomy of LPN. The manuscript is interesting and has merit. I only have a few comments:

Q1: Can the authors provide a better US image of the point where motor branch emerges from LPN? Also please mark other anatomical structures in the US image and add some details about US machine/probe in the Figure caption.

R2: We thank the reviewer for the suggestion. We included a more detailed figure (Figure 1) as suggested and a second figure (Figure 3) showing the US visualization after methylene blue injection. We also indicated anatomical structures and technical information.

Q2: How many times did authors perform this procedure? If this is a case report, the language should be written in past tens.

R2: We thank the reviewer for the valuable comment. In the present work we aimed at describing the modified procedure exclusively. We are planning to present data on large number of our cases in order to show other reached endpoints in future works.

Q3: Are there any limitations of using methylene blue? Can it obstruct the view of a surgeon if structures are colored blue? In our experience using 0,5% methylene blue, it can soil the tissue quite significantly, obstructing the clear view of anatomy. Can it be potentially locally toxic?

R3: We thank the reviewer for pointing this out. In our experience the use of Methylene Blue did not constitute an obstacle to surgeon view due to tissue soiling. We did not experience any local adverse events linked to MB use. Very few reports of methylene blue local toxicity exist and these are usually following more demolitive surgeries such as breast and colon using higher doses and different dilutions (see, for reference: Bužga et al. 2022. doi: 10.1093/toxres/tfac050). We included this aspect in the discussion.

Round 2

Reviewer 2 Report

Comments and Suggestions for Authors

The authors have improved the quality and presentation of their manuscript. I think it is suitable for publication as a technical note/communication.